# Convergent High O_2_ Affinity but Distinct ATP-Mediated Allosteric Regulation of Hemoglobins in Oviparous and Viviparous *Eremias* Lizards from the Qinghai-Tibet Plateau

**DOI:** 10.3390/ani14101440

**Published:** 2024-05-11

**Authors:** Peng Pu, Zhiyi Niu, Ming Ma, Xiaolong Tang, Qiang Chen

**Affiliations:** 1School of Biological and Pharmaceutical Engineering, Lanzhou Jiaotong University, Lanzhou 730070, China; pupeng@lzjtu.edu.cn; 2Department of Animal and Biomedical Sciences, School of Life Sciences, Lanzhou University, No. 222 Tianshui South Road, Lanzhou 730000, China; niuzhy20@lzu.edu.cn (Z.N.); mma2020@lzu.edu.cn (M.M.); tangxl@lzu.edu.cn (X.T.)

**Keywords:** lizard, high-altitude adaptation, hemoglobin, O_2_ affinity, allosteric regulation

## Abstract

**Simple Summary:**

This study investigates the functional adaptation and underlying molecular mechanisms of hemoglobins (Hbs) in the two species of *Eremias* lizards dwelling on the Qinghai-Tibet Plateau. By measuring O_2_ equilibrium curves of purified Hbs at different pH and temperature in the absence and presence of ATP and/or Cl^−^, the study found that the high-altitude populations of the two species of *Eremias* lizards exhibit convergent high Hb-O_2_ affinity compared to the respective lowland counterparts while demonstrating distinct ATP-mediated allosteric regulation. Hbs of the highland *E. argus* showed high ATP sensitivity and ATP-dependent strong Bohr effect compared to *E. multiocellata*. The underlying mechanisms of these functional variations may be attributed to the varying β2/β1 globin ratios, combined with substitutions on the β2-type globin, as suggested by Hb isoform identification and structural analysis of tetrameric Hbs. In addition, Hbs of these *Eremias* lizards have similarly low temperature sensitivities and relatively high Bohr effects at lower temperatures, which could minimize the impact of temperature fluctuations on Hb-O_2_ affinity and facilitate the release of O_2_ in the cold extremities at low temperatures.

**Abstract:**

The functional adaptation and underlying molecular mechanisms of hemoglobins (Hbs) have primarily concentrated on mammals and birds, with few reports on reptiles. This study aimed to investigate the convergent and species-specific high-altitude adaptation mechanisms of Hbs in two *Eremias* lizards from the Qinghai-Tibet Plateau. The Hbs of high-altitude *E. argus* and *E. multiocellata* were characterized by significantly high overall and intrinsic Hb-O_2_ affinity compared to their low-altitude populations. Despite the similarly low Cl^−^ sensitivities, the Hbs of high-altitude *E. argus* exhibited higher ATP sensitivity and ATP-dependent Bohr effects than that of *E. multiocellata*, which could facilitate O_2_ unloading in respiring tissues. *Eremias* lizards Hbs exhibited similarly low temperature sensitivities and relatively high Bohr effects at lower temperatures, which could help to stably deliver and release O_2_ to cold extremities at low temperatures. The oxygenation properties of Hbs in high-altitude populations might be attributed to varying ratios of β2/β1 globin and substitutions on the β2-type globin. Notably, the Asn12Ala in lowland *E. argus* could cause localized destabilization of the E-helix in the tetrameric Hb by elimination of hydrogen bonds, thereby resulting in its lowest O_2_ affinity. This study provides a valuable reference for the high-altitude adaptation mechanisms of hemoglobins in reptiles.

## 1. Introduction

High-altitude environments of the plateau provide an ideal laboratory for studying the adaptation mechanisms of animals to hypobaric hypoxia and low ambient temperature. The world’s highest-dwelling mammal is reported to be a specimen of the yellow-rumped leaf-eared mouse (*Phyllotis xanthopygus rupestris*), which was captured at the summit of Llullaillaco at an altitude of 6739 m [1]. The ability to regulate O_2_ tensions in systemic blood and respiring tissues was weaker for reptiles and amphibians because of the incomplete separation of systemic and pulmonary circulation [2]. However, the natural habitats of reptiles can extend to extremely high altitudes. The highest-altitude reptile in the world (*Liolaemus* aff. *tacnae*) was recently recorded at 5400 m elevation in Peru [3]. The red-tailed toad-head lizard (*Phrynocephalus erythrurus*) native to the Qinghai-Tibet Plateau (QTP) inhabits between 4500 and 5300 m in elevation [4]. The fact that high-altitude reptiles can survive and prosper at extremely high elevations indicates that they have evolved unique mechanisms to adapt to severe hypoxia and low temperatures.

Air-breathing vertebrates well adapted to high-altitude environments can adjust their respiratory and cardiovascular systems to maintain an adequate O_2_ supply to respiring tissues [5,6,7]. High Hb-O_2_ affinity could maximize O_2_ extraction from the lungs under hypoxia for animals that are endemic to plateaus. On the other hand, the efficiency of O_2_ unloading from Hb into aerobic tissues mainly depends on the sensitivity of Hb to allosteric effectors (Cl^−^, H^+^, CO_2_, and organic phosphates). The main organic phosphates are 2,3-diphosphoglycerate (DPG), inositol pentaphosphate (IP5), and adenosine triphosphate (ATP) in the erythrocytes of mammals, birds, and reptiles, respectively. Amphibian erythrocytes contain both DPG and ATP as allosteric effectors. Numerous studies have shown that high-altitude mammals and birds generally have a significantly higher Hb-O_2_ affinity compared to their low-altitude relatives or populations, such as the bar-headed goose [8,9,10,11], high-altitude passerine birds [12], high-altitude deer mice [13,14], Tibetan antelope [15], and plateau zokor (*Eospalax baileyi*) [16]. The high Hb-O_2_ affinities are the consequence of high intrinsic Hb-O_2_ affinity and/or reduced allosteric effector sensitivity. The underlying molecular mechanisms of high intrinsic Hb-O_2_ affinity are mainly attributed to the structural effects of specific amino acid substitutions. As found in the Hbs of bar-head geese and plateau zokors, substitutions located at the (αβ)2 interface could eliminate the T-state-stabilizing hydrogen connection on the interface between (α1β1/α2β2) or inside (α1/β1 and α2/β2) the two semirigid dimers, facilitating the conformational transition from the T-state to the R-state during Hb oxygenation and resulting in an increase in the intrinsic Hb-O_2_ affinity [10,11,16]. Amino acid substitutions located on the β chain heme pockets could fine-tune conformation of the pockets and subsequently reduce the steric hindrance of O_2_ binding, which could be another mechanism of the high intrinsic Hb-O_2_ affinity [14,16,17]. In addition, reduced allosteric effector sensitivity could also lead to high Hb-O_2_ affinity due to substitutions at binding sites of allosteric effectors, as exemplified by the Andean frog (*Telmatobius peruvianus*) and Andean llamas [18,19,20]. It has been reported that evolutionary adjustments in Hb function are also attributed to epistatic interactions between different substitutions [13]. Most of these studies focused on mammals, birds, and a few amphibians. However, little attention has been given to the high-altitude adaptation of Hb function and structure in reptiles despite their extremely high distribution, lack of physiological thermoregulation, and incomplete circulatory system.

The O_2_ affinity of Hb decreased with increasing temperature due to the exothermic nature of Hb oxygenation. A substantial temperature effect impairs O_2_ delivery to cold limbs and extremities in mammals living in cold environments [21]. Several polar and alpine mammals have been shown to possess Hbs with low temperature sensitivity [16,21,22,23]. Regional heterothermy may not exist both in ectothermic reptiles and amphibians due to a lack of physiological thermoregulatory capacity. However, our previous study revealed that Hbs of both high- and low-altitude Asiatic toads (*Bufo gargarizans*) exhibit low temperature sensitivity, which could help to reduce the temperature-induced fluctuations in Hb-O_2_ affinity [24]. Whether the same is true in alpine reptiles remains to be investigated.

*Eremias argus* (oviparous) and *Eremias multiocellata* (viviparous) are two of the eight *Eremias* (Family Lacertidae) species in China [4]. A molecular phylogeographical study based on 16S rRNA mitochondrial DNA revealed that *E. argus* clustered with *E. brenchleyi* on a monophyletic clade as the sister group of *E. multiocellata* [25]. The altitude distribution of the two *Eremias* lizards ranges from 0 to 3500 m in a broad dimension from the Eastern Plain to the QTP in China [4]. This provides a valuable opportunity to study the high-altitude adaptation of the two closely related lizards by comparing their high- and low-altitude populations. Recent comparative genomics analyses revealed that *E. argus* populations endemic to high altitudes possess many novel genomic regions under strong selective sweeps, and genes embedded in those regions are mainly associated with energy metabolism and DNA damage repair pathways [26]. However, high-altitude adaptation in the oxygen transport system has not been studied in the two *Eremias* lizards dwelling on the QTP.

The present study aimed to characterize the Hb isoform composition in erythrocytes, the oxygenation properties of Hbs, and the underlying genetic basis in high-altitude *E. argus* and *E. multiocellata* living on the QTP. We purified Hbs from hemolysates of high- and low-altitude *E. argus* and *E. multiocellata*, and measured intrinsic Hb-O_2_ affinity and their sensitivity to allosteric effectors (H^+^, Cl^−^ and/or ATP) and temperature. We also analyzed Hb isoform diversity and sequenced the coding DNA sequence data for the full complement of *α*- and *β*-type globin genes from *E. argus* and *E. multiocellata* to elucidate the mechanism underlying the observed functional properties.

## 2. Materials and Methods

### 2.1. Ethics Statement

All experiments were carried out according to the principles from the China Council on Animal Care, and approved by the Ethics Committee of School of Life Sciences, Lanzhou University (protocol code EAF2021026 and date of approval: 1 April 2021). Every effort was made to minimize the numbers used and any suffering experienced by the animals in the experiment.

### 2.2. Sample Collection

High-altitude *E. argus* (*n* = 6, average of 3.71 g) were collected at Shazhuyu township (36°17′06″ N, 100°35′56″ E, 2860 m), Qinghai Province, China. Low-altitude *E. argus* (*n* = 6, average of 3.76 g) were collected at Xingtai city (36°58′47″ N, 114°26′28″ E, 89 m), Hebei Province, China. High-altitude *E. multiocellata* (*n* = 6, average of 3.93 g) were collected at Tianzhu Tibetan Autonomous County (37°18′16″ N, 103°10′30″ E, 2837 m), Gansu Province, China. Low-altitude *E. multiocellata* (*n* = 6, average of 4.56 g) were collected at Lanzhou City (36°18′43″ N, 103°51′44″ E, 1728 m), Gansu Province, China. All lizards are adult males and were collected in May or August 2019 before or after the breeding season. The geographical location and climate data of the four sampling sites are shown in Appendix A. The climate data for the years from 1981 to 2010 were obtained from the Chinese Climatic Data Centre (http://data.cma.cn, accessed on 5 August 2021). 

Blood samples were drawn from the aortic arch directly using a heparinized glass capillary tube after the lizards were anesthetized with ether. Approximately 20–30 µL of blood sample was obtained from each lizard and then centrifuged at 4 °C (3000× *g*, 10 min) to obtain packed red blood cells. Liver samples were also harvested for cDNA cloning and sequencing of globins. All the samples were flash-frozen with liquid nitrogen and stored at −80 °C for subsequent experiments. Every effort was made to minimize the number of animals used and any suffering experienced by the animals during the experiments. 

### 2.3. RT-PCR and Sequencing of α and β Globins

To investigate the genetic variation of hemoglobin in high- and low-altitude *E. argus* and *E. multiocellata*, we cloned and sequenced the adult-expressed *α*- and *β*-type globin genes from lizards. Total RNA was extracted from liver samples (40–100 mg) using RNAiso Plus reagent (Takara, Dalian, China). Then, we eliminated residual genomic DNA and amplified full-length cDNAs of adult-expressed genes using a PrimeScript^TM^ RT reagent kit with gDNA Eraser (Takara, Dalian, China). We designed paralog-specific primers (Appendix A) using 5′ and 3′ sequences according to the annotated globin genes in the genome assemblies of the common wall lizard (*Podarcis muralis*) [27]. We cloned all target globin genes with the reverse-transcription (RT)-cDNA as the template using 2× Accurate Taq Master Mix (dye plus) (Accurate Biotechnology Co., Changsha, China). The target products were then connected with the pMD^TM^18-T vector (Takara, Dalian, China) and further transformed into DH5α-competent cells (Takara, Dalian, China). At least three positive clones per gene were sequenced (Sangon, Shanghai, China) to ensure the accuracy of the sequencing results. All new sequences have been submitted to the GenBank database under accession OL804548-OL804565.

### 2.4. Sequence Alignment and Phylogenetic Analysis

Nucleotide sequences of the globin genes were conceptually translated into amino acid sequences using MEGA 11 [28]. We compared our newly generated sequence data with the adult-expressed *α*- and *β*-type globin genes of 12 species of lizards from seven families of Sauria. The globin sequences of the remaining species were obtained from public databases or annotated from genome assemblies (Appendix A). 

It needs to be pointed out that the names of β1 and β2 in anole were inaccurate in the previous study [29]. We aligned the β^II^ in their study to the nr database with the BLASTP of NCBI and found that β^II^ was completely consistent with the β1 annotated in the anole genome [30]. The re-annotation of the anole genome by Lu S. (2017) also confirmed that β^I^ and β^II^ should be β2 and β1, respectively [31]. In addition, the β1- and β2-type globins of *Lacerta agilis* also had the same naming problem as anole. BLASTP results found that its β1 was most similar to β2 of *P. muralis*, while its β2 was most similar to β1 of *Z. vivipara* and *P. muralis*. Therefore, we renamed the β1 and β2 of the anole and *L. agilis* in this study based on the BLASTP results.

It has been revealed that the *α^E^*-, *α^D^*-, and *α^A^*-globin genes diverged through duplication events before the radiation of tetrapods [29,32,33]. Thus, the homologous sequences from human (*Homo sapiens*) and chicken (*Gallus gallus*) were included for alignment of amino acid sequences and reconstruction of the phylogenetic relationships of α and β globin of *Eremias* lizard. The amino acid sequences of α- and β-type globins were aligned using muscle implementation in MEGA 11. Subsequently, the maximum likelihood phylogenies for α-type and β-type were estimated using LG + G + I and LG + G models of amino acid substitution with five different site categories, respectively. The support rate of branch nodes was evaluated by 1000 bootstrap pseudoreplicates.

### 2.5. Hb Purification and Isoform Identification

The frozen red blood cells of six individuals from each population were incubated on ice for 20 min after adding a 5-fold volume of ice-cold 10 mmol/L of Hepes buffer (pH7.8, 0.5 mmol/L of EDTA). Then, hemolysates were centrifuged (9000× *g*, 10 min at 4 °C) to remove membranes and cellular debris, and the supernatants from the six individuals were pooled together for further purification. The ÄKTA Pure chromatography system (GE Healthcare, Chicago, IL, USA) and IexCap Q 6FF 5 mL column (Smart-lifesciences, Changzhou, China) were used to remove miscellaneous proteins and endogenous organic phosphates in supernatants. Before and after the injection of the supernatants, the anion exchange column was equilibrated with 20 mmol/L of Tris·HCl (pH8.7, 0.5 mmol/L of EDTA) and then eluted with a linear gradient of 0–400 mmol/L of NaCl at 1 mL/min flow rate to obtain mixtures of hemoglobin isoforms (Hbs). The mixtures were desalted by dialyzing against three changes of a 200-fold volume of 10 mmol/L of Hepes buffer (pH7.6, 0.5 mmol/L of EDTA) at 4 °C. The Hb isoform (isoHb) compositions were verified using isoelectric focusing (IEF) on polyacrylamide gels in the pH range of 3–10 (DYCP-37B, Liuyi Biotechnology, Beijing, China). The final purified Hbs were concentrated to ~1.3 mmol/L of heme by ultrafiltration using Amicon^®^ Ultra-4 Centrifugal Filter Units fitted with a 10 kDa cutoff filter (Millipore, Nantong, China) and then stored at −80 °C in aliquots.

To identify whether the sequenced globins were all expressed in hemolysates, Hb subunits of native hemolysates were separated on a 15% SDS-PAGE gel, digested with trypsin and determined by liquid chromatography–tandem mass spectrometry (LC–MS/MS) using Orbitrap Fusion Lumos MS (Thermo Fisher Scientific, Waltham, MA, USA) coupled online to an EASY-nLC 1200 system in a data-dependent mode (DDA) according to the previously described method [34]. The Hb isoform compositions of erythrocytes from *Eremias* lizards were further measured using RP-HPLC (Bio-Bond C4 column, 5 μm, 250 × 4.6 mm, DIKMA, Beijing, China) and an ultrahigh-resolution time-of-flight mass spectrometer (MaXis 4G, Bruker-Daltonics, Billerica, MA, USA) using the methods and parameters described by Lu et al. [35].

### 2.6. Measuring O_2_ Equilibrium Curves

O_2_ equilibrium curves of the purified Hbs were measured using a homemade modified diffusion chamber as previously described [11,36,37,38]. Purified Hbs (0.3 mmol/L of heme) were diluted in 0.1 mol/L of Hepes buffers in the absence (stripped) and presence of Cl^−^ (added as 0.1 mol/L of KCl) and/or ATP (7.5 mmol/L, 100-fold molar excess over tetrameric Hbs). Absorbance at 436 nm of the working Hbs solutions (≈4 µL) was continually monitored under the 100% humidified gas mixture, in which O_2_ tension ($P_{O_{2}}$, Torr) was stepwise increased by mixing ultrapure N_2_ and air. A_0_ and A_100_ are the absorbances at zero and full O_2_ saturation equilibrated with ultrapure N_2_ and atmospheric air, respectively. Fractional saturation ($S_{O_{2}}$) under the corresponding $P_{O_{2}}$ was calculated by $S_{O_{2}}$ = (A − A_0_)/(A_100_ − A_0_). O_2_ equilibrium curves were measured in 0.1 mol/L of pH7.4 and 7.8 Hepes buffers at 25 °C and 37 °C under different allosteric conditions (stripped, Cl^−^ and/or ATP present) to calculate the Bohr effect, enthalpy of oxygenation, and anionic cofactor sensitivities. The experimental temperature was based on the ranges of body temperatures (24.1–37.2 °C) at which adult lizards maintained 80% of the maximum sprint speed for *E. multiocellata* [39]. 

At least six technical repeats were performed for each experimental condition. The pH of the working Hbs solutions used for O_2_ equilibrium experiments was adjusted with NaOH to as close to 7.4 and 7.8 as possible and then precisely measured with an InLab micro pH electrode equipped with a SevenCompact pH/Ion Meter S220 and an ATC temperature probe (Mettler Toledo, Greifensee, Switzerland) after the samples were brought to the same temperature as that used in the experiments. To further assess the effect of allosteric binding of ATP on the Hb-O_2_ affinity of lizard Hbs, we measured O_2_ equilibrium curves with ATP concentrations in the range of 0–7.5 mmol/L in the absence of KCl at 0.3 mmol/L [heme] at 37 °C in 0.1 mol/L of Hepes buffer (pH7.4).

The dose–response curves of ATP on P_50_ were generated by fitting a hyperbolic function to the data, and the apparent affinity constant was calculated as the ATP concentration at which the increase in logP_50_ was half the maximum [40].

P_50_ is the $P_{O_{2}}$ at half-saturation, while n_50_ is the Hill cooperativity coefficient. We fitted the Hill’s sigmoidal equation $\left[S_{O_{2}}={P_{O_{2}}}^{n_{50}}/\left({P_{50}}^{n_{50}}+{P_{O_{2}}}^{n_{50}}\right)\right]$ to 5–10 saturation data (O_2_ saturation range ∼0.1–0.9) to estimate the values of P_50_ and n_50_ for each condition. 

Nonlinear sigmoidal Hill fitting (*r*^2^ > 0.99) was performed on GraphPad Prism 8 (San Diego, CA, USA). The Bohr factor (Φ) was used to quantify the Bohr effect and was calculated as the slope of linear plots of logP_50_ as a function of pH in the range of pH7.4–7.8 (Φ = ΔlogP_50_/ΔpH). The overall change in enthalpy of oxygenation (ΔH, kJ/mol O_2_), which is the heat liberated upon oxygenation, was calculated by the van’t Hoff equation ΔH = 2.303R(ΔlogP_50_)/Δ(1/T) and used to indicate the temperature sensitivity of Hbs. R is the gas constant (8.314 J/K/mol O_2_), and T is the absolute temperature in Kelvin. The final ΔH values were calibrated by excluding the solution heat of O_2_ (ΔH^sol^ ≈ −12.6 kJ/mol O_2_) [41].

### 2.7. Homology Modeling and Molecular Dynamics Simulation of Tetrameric Hb

According to our sequencing results, αA, αD, and β1 globins of high- and low-altitude populations are the same or only different at one site, and β2 globins of the two high-altitude populations are the same. The amino acid substitutions of Hb globins between high- and low-altitude populations mainly occurred on the β2-type globin for both *E. argus* and *E. multiocellata*. So, we first constructed the tetramer model (T-state) composed of β2 globin with all α-type globins (αAβ2-AB2, αD2β1-D2B1, αD2β2-D2B2) using Modeller 10.1 software [42]. The crystal structure of Turkey *(Meleagiris gallopova)* deoxyhemoglobin at 2.3 Å (3K8B.pdb) was selected as the template for homology modeling because of its highest percent identity (69.18%) with *Eremias* β2 globin. Molecular dynamics simulations were further performed on the Hb models to explore the quaternary structural properties of deoxyhemoglobin under simulated physiological conditions according to the previous method using NAMD 2.15 software [35,43]. The output dcd trajectory files of 10.5 ns molecular dynamics simulations were finally analyzed using VMD 1.9.4 software and its Plugs to explore the structural properties of different Hb models at the last 4 ns. 

### 2.8. Statistical Analyses

Curve fitting and statistical analysis were performed using GraphPad Prism 8.4.3 (GraphPad Software, San Diego, CA, USA) and IBM SPSS Statistics 26, respectively. A schematic diagram of the tetrameric Hb spatial structure was drawn via VMD. The normality and homogeneity of the P_50_ values were checked, and then the one-way ANOVA and independent sample *t*-test were used to detect significant differences (*p* < 0.05) between populations of the same and different species, respectively. The significant differences between the stripped state and conditions for adding KCl and/or ATP under the same temperature and pH were analyzed using the LSD (equal variances) and Games–Howell (unequal variance) method of post hoc multiple comparisons of one-way ANOVA. P_50_ and n_50_ values are presented as mean ± SE.

## 3. Results

### 3.1. Sequence Variation, Phylogenetic Relationships, and isoHb Compositions

Amino acid sequence alignment revealed that *E. argus* and *E. multiocellata* typically possess a full repertoire of αA-, αD-, β1-, and β2-type globins similar to most phylogenetically related lizards (Figure 1). The phylogenetic relationships based on the amino acid sequences showed that the αA- and αD-type globins of lizards are orthologous and sister to the αE-globin clade of additional amniote outgroup taxa (Figure 2). The phylogenetic analysis of β-type globins indicated that the β1- and β2-type globins of Sauria lizards were products of a duplication event that occurred before the divergence of Sauria lizards, as the β1 globins of all lizards are sister to the β2 globin of all lizards (Figure 2). The αA-, αD-, β1-, and β2-type globins of Lacertidae lizards (*Eremias*, *L. Agilis*, *Z. vivipara*, and *P. muralis*) are clustered on a clade, and all globins of *Eremias* are sister to those of *L. Agilis* (Figure 2). Surprisingly, high- and low-altitude *E. multiocellata* additionally expressed a distinct αD-type globin (αD2), which grouped in the same clade with αD1 (Figure 2). The second position of the N-terminus in *E. multiocellata* αD2-globin is Met compared with the αD of other species. As shown in Figure 1, the amino acid sequences of the αA globin are the same in *E. argus* and *E. multiocellata*; one substitution was observed in the αD1 globin between high- and low-altitude *E. argus* (Met73Leu) and in the β1 globin between high- and low-altitude *E. multiocellata* (Gln58Pro). The β1 globin of *E. argus* is heterozygous at site 56 (Ala/Ser). The amino acid sequences of the β2 globin are the same in the high-altitude *E. argus* and *E. multiocellata* (Figure 1), and four substitutions occurred on the β2 globin of the low-altitude *E. multiocellata* (Leu11Ile, Gly16Ser, Gly43Ala, Leu71Phe); seven substitutions occurred on the β2 globin of the low-altitude *E. argus* (Thr4Ser, Ala12Asn, Met20Ile, Thr25Gly, Ser27Thr, Ile54Val, Gly83Ala).

There are three major Hb isoforms with isoelectric points approximately equal to pH8.0, pH7.8, and 7.5 in the purified Hbs mixture of both *E. argus* and *E. multiocellata* (Appendix A). The results from the Orbitrap Fusion Lumos MS confirmed that all the sequenced α- and β-type globins were expressed in the hemolysates of the four populations of *Eremias* lizards. The results from RP-HPLC and ultrahigh-resolution time-of-flight mass spectrometer showed approximately equal amounts but quite different shapes of globin peaks in the two species of *Eremias* lizards (Figure 3). However, only αA, β1, and β2 globins were verified by similar molecular weights (MWs) to the theoretical values in these populations (Appendix A and Figure 3). Conversely, αD globin was not detected in any of the populations, as no peaks with MWs between 15,972.32 Da and 15,999.37 Da were detected.

αA globins were the first to be eluted, β1 globins were the last to be eluted in all studied populations, and β2 globins were the penultimate globins eluted in two populations of *E. argus*, while the antepenultimate globins were eluted in two populations of *E. multiocellata* (Figure 3). The ratio of β2/β1 globin was the highest in high-altitude *E. multiocellata* (~2.25), the same in two populations of *E. argus* (~1.38), and the lowest in low-altitude *E. multiocellata* (~1.23) based on the analysis of the HPLC peak using ImageJ 1.54g software. However, there are additional peaks whose MWs did not match our sequenced globins. Two peaks with MW values of 14,728.5000 Da and 14,938.8750 Da, and three peaks with MW values of 14,924.8425 Da, 14,939.0625 Da, and 14,938.8750 Da, were identified in high-altitude and low-altitude *E. argus*, respectively. Two distinct peaks, with similar interpopulation elution times and MW values, were identified in high-altitude (MW = 14,862.7500 Da, 16,238.6250 Da) and low-altitude (MW = 14,897.8125 Da, 16,240.3125 Da) *E. multiocellata*.

### 3.2. Convergent High Hb-O_2_ Affinity but Distinct ATP-Mediated Allosteric Regulation of Hbs in Eremias Lizards

For both *E. argus* and *E. multiocellata*, the O_2_ equilibrium curves were left-shifted, and the P_50_ values were significantly lower in the Hbs of the high-altitude population than in those of the low-altitude population under all experiment conditions (F_1,11_ = 32.76–1173.98 for comparisons among populations of the same species, *p* < 0.01, Figure 4 and Figure 5, Appendix A). The P_50_ values for the Hbs of the four populations significantly increased in the presence of KCl and/or ATP compared with those in the stripped state (F_1,11_ = 51.75–47,310.77, *p* < 0.05, Appendix A). In the stripped state, P_50_ values of high-altitude *E. argus* (2860 m) were the lowest. However, the interspecies differences in the P_50_ values of the two high-altitude populations in the stripped state and the presence of KCl were small, as were those of the two low-altitude populations (Figure 4). In the presence of ATP and ATP + Cl^−^, P_50_ values of *E. multiocellata*-H (2837 m), *E. multiocellata*-L (1728 m), and *E. argus*-L (89 m) were greatly decreased with the increase in elevation; however, high-altitude *E. argus* (2860 m) exhibited significantly higher P_50_ values compared to high- and low-altitude *E. multiocellata* (Figure 5). This is attributed to the anion allosteric effector sensitivity, which was expressed as the logP_50_ difference between adding Cl^−^ and/or ATP and the stripped state (Table 1). The Hbs of high-altitude *E. argus* possessed the highest ATP and ATP + Cl^−^ sensitivities, while these sensitivities were relatively low for high- and low-altitude *E. multiocellata* (Table 1). Furthermore, the ATP and ATP + Cl^−^ sensitivities of Hbs increased with decreasing temperature and pH in *E. argus* but not in *E. multiocellata*. Hbs of the four populations exhibit dramatically lower Cl^−^ sensitivities than ATP and ATP + Cl^−^ sensitivities. O_2_ binding was cooperative under all conditions (n_50_ = 1.89–3.08; Appendix A), reflecting a normal allosteric T–R shift upon oxygenation.

Dose–response curves (Appendix A) showed that the maximum ATP-induced logP_50_ value appeared at 2.0 mmol/L of ATP (approximately 27-fold molar excess over tetrameric Hbs), which indicated that the ATP concentration (7.5 mmol/L, 100-fold molar excess over tetrameric Hbs) used in our experiment was sufficient to fully reflect its effects on the Hb-O_2_ affinity. *E. argus* Hbs had a higher affinity for ATP than *E. multiocellata* Hbs, with estimated apparent binding constants of 0.19, 0.18, 0.31, and 0.27 mmol/L for *E. argus*-H, *E. argus*-L, *E. multiocellata*-H, and *E. multiocellata*-L, respectively. This result was consistent with the difference in ATP sensitives (ΔlogP_50(ATP-stripped)_) between the two species at 37 °C (Table 1).

The O_2_-binding curves shifted to the right when the pH decreased from 7.8 to 7.4 (Figure 4). The Bohr effect was indicated by the magnitude of Bohr factors (Φ = ∆logP_50_/∆pH), which are equal to the slopes of the linear plots in Figure 6. The Bohr factors of Hbs in high-altitude and low-altitude populations were similar under each allosteric condition for both species (Table 2). In addition, the Bohr factors at 25 °C were greater than those at 37 °C for the four populations in the stripped state and the presence of ATP (Figure 6). For the interspecific comparison, the Bohr factors were similar in *E. argus* and *E. multiocellata* Hbs in the absence of ATP (stripped state and only KCl added). In the presence of ATP, Bohr factors of *E. argus* Hbs markedly increased, but not for *E. multiocellata* Hbs, resulting in a stronger Bohr effect of *E. argus* Hbs than that of *E. multiocellata* Hbs in the presence of ATP (except for *E. multiocellata*-H when ATP was solely added, Table 2).

### 3.3. Low Temperature Sensitivity of Hbs in Eremias Lizards

The O_2_-binding curves shifted to the right when the temperature increased from 25 to 37 °C (Figure 4). As shown in Table 3, the temperature sensitivity of purified Hbs was quantified using the calibrated overall change in enthalpy (ΔH, kJ/mol O_2_). The ΔH at pH7.8 was higher than that at pH7.4 for the Hbs of the four populations under each allosteric condition except for *E. argus* in the presence of Cl^−^. For the stripped state, the ΔH of high-altitude population Hbs was higher than that of low-altitude population Hbs for both *E. argus* and *E. multiocellata*. The presence of Cl^−^, ATP, and ATP + Cl^−^ decreased the ΔH except for low-altitude *E. multiocellata* Hbs in the presence of ATP. Under the simulated physiological conditions in which both ATP and Cl^−^ were added, ΔH values were consistently low at pH7.4 (−4.12 to 5.97 kJ/mol O_2_) and pH7.8 (−9.41 to −14.52 kJ/mol O_2_).

### 3.4. Structural Properties of Tetrameric Hbs in T-State

The average backbone RMSD of AB2 (αAβ2) and DB2 (αD1β2 and αD2β2) Hb models of the four populations reached equilibrium in the last 5 ns of the simulations (Appendix A). The analysis of hydrogen bond interactions showed that side-chain -NH2 of 12Asn on the β subunit can form hydrogen bonds with 75Ile and/or 76Lys on the adjacent helix both in the AB2 and DB2 models of high-altitude *E. argus* and *E. multiocellata*, and low-altitude *E. multiocellata*. However, these hydrogen bonds were lost when 12β2 was Ala in the lowest *E. argus* (Figure 7). As for the four different amino acid substitutions of low-altitude *E. multiocellata* β2 globin in the AB2 and DB2 models, they only form hydrogen bonds with neighboring amino acids like the high-altitude *E. multiocellata*.

## 4. Discussion

### 4.1. Phylogenetic Relationship of Beta Globins in Sauria Lizards

The phylogenetic relationships of the α globin genes of Sauria lizards and outgroups (Figure 2A) estimated using our globin sequencing results (Figure 1) were consistent with previous studies. The squamates have retained *αD* and *αA* and lost the *αE* globin genes [32,33,44]. With respect to the β-globin, previous studies have suggested that each of the major groups of Sauropsids (birds, crocodilians, testudines, and squamates) evolved distinct β-globin repertoires via repeated rounds of lineage-specific gene duplication [33,44]. However, this phenomenon has not been clearly verified in Sauria lizards. In addition to the high similarity of β1- and β2-globin, several limitations may account for the unresolved evolutionary pattern of the *β*-globin gene in lizards [44]. Firstly, the lizards used in the previous analysis only come from only a few families of Sauria. Secondly, the β-globins used in the previous analysis were inaccurately annotated and did not contain the completed repertoires of β globin (lack β1 or β2). As described in the Appendix A (Appendix A), the name of the β1 and β2 of anole and *L. agilis* were inaccurate, *P. erythrurus* and *P. przewalskii* only expressed β1 and β2 globins, which was corrected in the followed study [31]. In this study, we collected more complete repertoires of β-globin from lizards and corrected some inaccurate annotations. Our estimated phylogenetic relationships showed that the adult-expressed β2 globins of lizards were nested within a clade containing β1 globins, and the ancestral β globins of the two lizard β-type globins were sister to those of chickens and humans (Figure 2B). This result suggested that the distinct β-globin repertoires of Sauria lizards may also have derived from lineage-specific duplication, similar to other lineages of Sauropsids [33,44].

### 4.2. The Physiological Significances of Oxygenation Properties of Eremias Lizards Hbs 

One of the primary findings of this study is that the Hbs of high-altitude *E. argus* and *E. multiocellata* both evolved genetically based high Hb-O_2_ affinity compared with their low-altitude populations. Given that the ATP and/or Cl^−^ sensitivities of Hbs were similar between high- and low-populations, the overall high Hb-O_2_ affinities of the high-altitude population for the two species were mainly attributed to their significantly elevated intrinsic Hb-O_2_ affinities. Significantly increased Hb-O_2_ affinity has been found in other high-altitude ectothermic reptiles and amphibians, such as the high-altitude red-tailed toad-head lizard (*P. erythrurus*) and Asiatic toad (*B. gargarizans*) from the QTP, and *Telmatobius* frogs and *Bufo spinulosus flavolineatus* from the Andes [18,24,35,45,46]. The evolved high Hb-O_2_ affinity could be the primary strategy for high-altitude reptiles and amphibians to ensure pulmonary O_2_ uptake under hypoxia in the incomplete systemic circulation.

The distinct ATP-mediated allosteric regulation of Hbs in the two closely related *Eremias* lizards may be associated with their different habitat preference [47]. *E. multiocellata* prefers open habitats that may increase O_2_ demand to evade predators. The high Hb-O_2_ affinity of high-altitude *E. multiocellata* under all allosteric conditions could ensure an adequate O_2_ supply. *E. argus* prefers close habitats that may minimize the impact of predators. The high ATP sensitivity and strong Bohr effect in the presence of ATP of high-altitude *E. argus* could then facilitate O_2_ unloading in systemic capillaries and therefore compensate for the side effect of its high Hb-O_2_ affinity on O_2_ unloading in tissues. The inherent adaptive increase in Hb-O_2_ affinity in high-altitude *E. multiocellata* may be compensated by other mechanisms, such as an increase in the tissue O_2_ diffusion capacity via increased muscle capillarization and plastic changes in Hb concentration and hematocrit (unpublished data). 

In the stripped state and the presence of Cl^−^, the Bohr effects were generally similar in the four populations’ Hbs (−0.15 to −0.25 at 25 °C, −0.08 to −0.18 at 37 °C). The presence of ATP could enhance the Bohr effects of high- and low-altitude *E. argus* Hbs (−0.41 to −0.45 at 25 °C, −0.30 at 37 °C), indicating that ATP could promote the binding of H^+^ to *E. argus* Hbs. However, this ATP enhancement was not found in *E. multiocellata* (Table 2). Similarly, in snake Hb, the alkaline Bohr effect is greatly enhanced in the presence of ATP [48,49,50], and this effect could result from ATP-induced polymerization of subunits [51]. In addition, the Bohr effects at 25 °C were greater than those at 37 °C for Hbs of all *Eremias* in the presence of ATP (Table 2, Figure 6). Given that the body temperature of ectotherm is determined by the ambient temperature, a higher Bohr effect at lower temperatures could facilitate O_2_ unloading to the metabolic tissue at low temperature. This is especially important for high-altitude *E. argus* and *E. multiocellata*, for which the mean ground and air temperatures during their active period (April to October) range from 2.3 to 18.6 °C (Appendix A). A thermoregulation study in the cold-climate *Phrynocephalus guinanensis* showed that the heating and cooling rates of limbs and tails were relatively faster than those of torsos [52]. This phenomenon may also exist in *Eremias* lizards, which have a body size similar to that of *Phrynocephalus* lizards. Thus, the relatively high Bohr effect of *Eremias* lizard’s Hbs at lower temperatures could promote the release of O_2_ in the limbs and tails when the ambient temperature drops rapidly, thereby enhancing their low temperature tolerance. Higher Bohr effects at lower temperatures were also found in high-altitude Asiatic toads from the QTP (Φ = −0.50 and −0.38 at 10 and 20 °C in the presence of ATP and KCl, respectively) [24]. However, this phenomenon is not exclusive to ectotherms and can also be found in human and horse Hb [53].

The temperature sensitivity of Hbs in *Eremias* lizards decreased with decreasing pH under almost all allosteric conditions (Table 3), indicating the endothermic binding of H^+^ to Hbs. The overall temperature sensitivities (on average −9.68 kJ/mol O_2_, Table 3) of Hbs in the two *Eremias* species were noticeably lower than those of Hbs in other species, such as temperature-sensitive human HbA (−50.7 kJ/mol O_2_), temperature-insensitive bovine Hb (−22.9 kJ/mol O_2_), deer mouse Hbs (on average −10.5 kJ/mol O_2_), and Asiatic toad Hbs (on average −16.57 kJ/mol O_2_) [23,24,54]. As mentioned above, the cooling rates of limbs and tails were relatively faster than those of torsos in cold-climate lizards. Therefore, when the ambient temperature drops rapidly, the low temperature sensitivity of Hbs in *Eremias* lizards could minimize the hindering effect of temperature-induced high Hb-O_2_ affinity on the transport and release of O_2_ to the limbs and tail. In addition, the body temperature of *Eremias* lizards is mainly determined by the ambient temperature, and the mean air and ground temperatures fluctuate greatly during different months throughout the year at the sampling sites (Appendix A), and the temperature difference between day and night is also great in the plateau environment. Thus, the low temperature sensitivity of *Eremias* lizard Hbs, especially under relatively low pH conditions, could minimize the temperature- and pH-induced fluctuations in Hb-O_2_ affinity during the transport of O_2_ to cold limbs and metabolic tissue and finally safeguard the tissue O_2_ supply [21].

Low Cl^−^ sensitivity (∆logP_50_ = 0.04–0.20) was found in the Hbs of high- and low-altitude populations in both species compared with the ATP sensitivity (∆logP_50_ = 0.30–0.62) (Table 1). Disregarding any other variations, this property will ensure O_2_ loading under hypoxia without the need for decreasing erythrocytic organic phosphate levels and the related allosteric regulatory capacity [18]. The drastically suppressed Cl^−^ sensitivity has been revealed in other reptiles and amphibians, such as *Phrynocephalus* lizards from China (unpublished data from the author), the aquatic Andean frog *T. peruvianus*, Asiatic toad (*B. gargarizans*), and black-spotted frog (*Pelophylax nigromaculatus*) [18,24,55,56]. We speculate that the anion-related allosteric regulation of hemoglobins in some reptiles and amphibians is primarily mediated by organic phosphates rather than chloride ions. More investigations into the oxygenation properties of Hbs in reptiles and amphibians are necessary to verify this hypothesis.

### 4.3. Molecular Mechanisms Underlying the Oxygenation Properties of Eremias Hbs

The molecular mechanisms underlying the high Hb-O_2_ affinity are not only related to evolutionary changes in globin sequences but are also influenced by Hb isoform diversity [57,58]. αD-type globin was detected in all hemolysates of *Eremias* lizards by Orbitrap Fusion Lumos MS but not by the time-of-flight mass spectrometry using products separated by RP-HPLC. These inconsistent results suggest that αD-type globin accounts for a small proportion of the hemolysate of *Eremias* lizards, while αA-type hemoglobin is the predominant isoform. Our results align with the Hb isoform differentiation observed in turtles and birds. In most turtles and birds, HbA composed of αA-type globin is the major isoform, and HbD composed of αD-type globin is the minor isoform [38,57,59,60]. However, the opposite pattern of isoHb differentiation was found in the green anole lizard (*Anolis carolinensis*) and South American rattlesnake (*Crotalus durissus*) [29,51]. In these species, HbD is the major isoform, and its O_2_ affinity is lower than that of HbA. Thus, Hb isoform diversity might be species-specific in squamate reptiles. In addition, a consistently higher O_2_ affinity of HbD than that of HbA in all examined turtles and birds was not verified in *Eremias* lizards because the individual Hb isoforms were not separated in this study, which should be resolved in further studies [61,62]. For globin sequence variation, the amino acid sequences of the αA globin were identical in the highland and lowland *E. argus* and *E. multiocellata*.

The oxygenation properties of Hbs in the studied *Eremias* lizards might largely be attributed to differences in β-globin. The relative abundance of β2 globin to β1 globin was the highest in high-altitude *E. multiocellata* (~2.25), followed by the two populations of *E. argus* (~1.38), and the lowest in low-altitude *E. multiocellata* (~1.23). For interpopulation comparison in *E. argus*, high O_2_ affinity of Hbs in the high-altitude population could be attributed to the seven substitutions in the β2 globin (Figure 1), as the β2/β1 ratio and amino acid sequence of the β1 globin were identical in the high- and low-altitude populations (Figure 1 and Figure 3). For interpopulation comparison in *E. multiocellata*, high O_2_ affinity of Hbs in the high-altitude population could be attributed to four substitutions on the β2 globin, the high relative abundance of the β2 globin, and one substitution on the β1 globin (Gln → Pro) (Figure 1 and Figure 3). In the comparison between the high-altitude populations of *E. argus* and *E. multiocellata*, the amino acid sequences of their β2 globins were identical, and the main differences were the β2/β1 globin ratio and eight species-specific amino acid variations in the β1 globin, which may result in the higher ATP sensitivities and ATP-dependent high Bohr effect in the Hbs of highland *E. argus* relative to those of *E. multiocellata*. In addition, because of their distinct ATP-dependent oxygenation properties, the O_2_ affinity of highland *E. argus* was higher than that of highland *E. multiocellata* in the absence of ATP but lower than that of highland *E. multiocellata* in the presence of ATP. Furthermore, additional peaks whose MWs cannot match our sequenced results may also be attributable to the differences in the oxygenation properties of the highland populations of the two species. These peaks may be unsequenced globins or products of the deamination or other forms of posttranslational modification of the sequenced globins. Studies on the isoform diversity of snake hemoglobins also showed that products of the β2 globin gene form the main isoHbs in the South American rattlesnake and leaf-nosed viper (*Eristicophis macmahonii*) snakes [51,63].

The lowest intrinsic Hb-O_2_ affinity of low-altitude *E. argus* Hbs may also be related to the localized destabilization of the E-helix secondary structure in Hb models composed of β2 globin with all α globins (Figure 7). The substitution of Asn to Ala at the 12 position on β2 globin in the lowland *E. argus* caused the elimination of hydrogen bonds formed between 12β2 and 75Ile and/or 76Lys on the adjacent E-helix compared with the other three *Eremias* lizards (Figure 7B,C). A similar structural mechanism was previously found in high-altitude American pika (*Ochotona princeps*), where the β62 Ala → Thr substitution could increase the rigidity of the E-helix, consequently leading to an increased oxygen affinity [17]. 

It is still not clear which amino acid residues are responsible for the different ATP sensitivities of the two closely related species of *Eremias* lizards since β1Val, β82Lys, and β143Arg are conserved in *Eremias* Hbs, except that the positively charged β2His is replaced by negatively charged β2Glu. Studies on fish Hbs have shown that substitutions of Glu or His at the position 2 in the β chain do not affect DPG sensitivity [64,65]. We speculate that the high ATP-binding affinities of the high-altitude *E. argus* Hbs could be attributed to the combined effects of different amino acid substitutions of the αD and β2 globins, coupled with isoform diversity in the β globins.

Cl^−^ binds to an α-chain site (between α1Val and α131Ser) and a β-chain site (between β1Val and β82Lys) in human HbA [66]. However, the polar 131Ser is replaced by nonpolar Val and electronegative Glu in αD1 and αD2 of *Eremias* Hbs (Figure 4A), respectively. In addition, the N-terminal residue 1Val of αD2-globin in *E. multiocellata* was replaced by Met (Figure 1A). These substitutions may result in the elimination of Cl^−^ binding sites in Hbs formed by αD1- and αD2-type globins. Similar mechanisms were found in the Andean frog (*T. peruvianus*) and Indian python (*Python molurus*) Hb [18,51]. However, it is uncertain whether the loss of Cl^−^ binding sites at the αD2 globin could cause the dramatically low Cl^−^ sensitivities of *Eremias* lizard Hbs, as the αD globins were not isolated by HPLC in this study. 

## 5. Conclusions

In this study, we investigated the oxygenation properties and underlying mechanisms of Hbs in two closely related species of *Eremias* lizards from the QTP compared with their low-altitude counterparts. Phylogenetic analysis suggested that the evolved distinct β-globin repertoires of Sauria lizards may also be derived from lineage-specific duplications. The significantly high overall and intrinsic Hb-O_2_ affinity of the highland *E. argus and E. multiocellata* Hbs can ensure efficient pulmonary O_2_ uptake under hypoxic conditions compared to lowland populations. The Hbs of high-altitude *E. argus* exhibit higher ATP sensitivities and stronger ATP-dependent Bohr effects than that of *E. multiocellata*, and the mechanism may be related to the differential abundance of β2 and β1 globins, as well as specific amino acid substitutions in the β2-type globin. These properties could compensate for the high Hb-O_2_ affinity and thus facilitate O_2_ unloading in respiring tissues for the high-altitude *E. argus*. Furthermore, Hbs of *Eremias* lizards Hbs exhibited low temperature sensitivities, which decreased with decreasing pH, and higher Bohr effects were observed at lower temperatures. Taken together, these characteristics could minimize the temperature- and pH-induced fluctuations in the Hb-O_2_ affinity and facilitate the transport and release of O_2_ to cold limbs and metabolic tissue at low temperatures. Our results could provide a valuable reference for the study of adaptation mechanisms of Hbs in other high-altitude reptiles.

## Figures and Tables

**Figure 1 animals-14-01440-f001:**
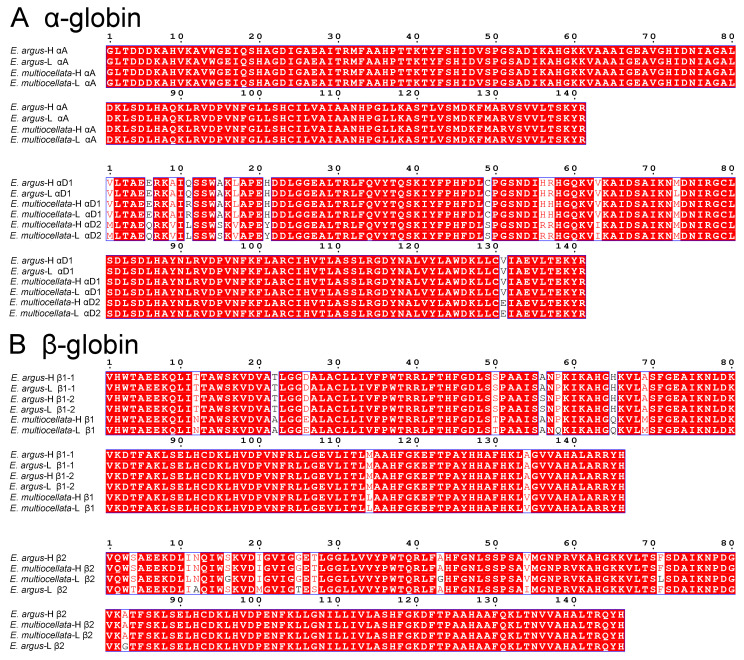
Alignment of amino acid sequences for the adult-expressed *α*- (**A**) and *β*-type globin genes (**B**) from high- and low-altitude *E. argus* and *E. multiocellata* performed on the ESPript 3.0 online tool.

**Figure 2 animals-14-01440-f002:**
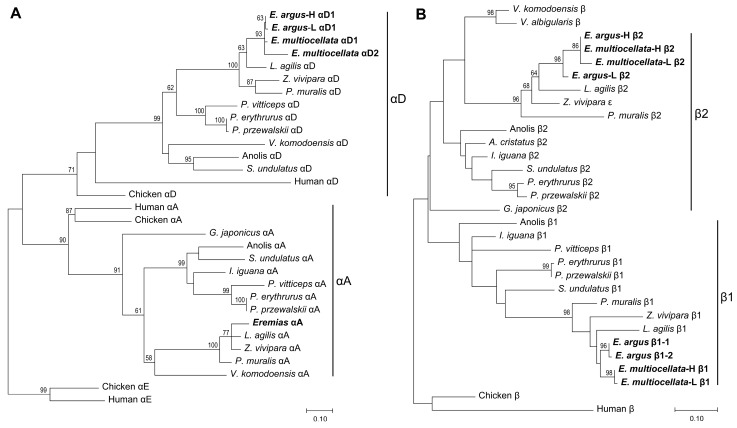
Maximum likelihood phylogenies of α- (**A**) and β-type globins (**B**) of Sauria lizards constructed by amino acid sequences, and the full set of adult-expressed α- and β-type globins from *E. argus* and *E. multiocellata* are highlighted with bold font. Bootstrap percentages are shown on relevant nodes to indicate the support values.

**Figure 3 animals-14-01440-f003:**
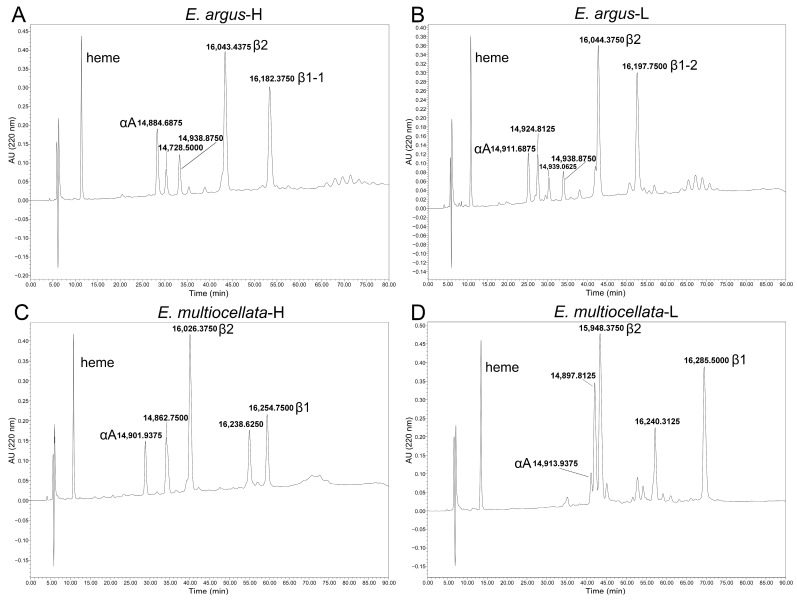
RP-HPLC chromatograms for erythrocyte hemolysates of *Eremias* lizards. Five and six globin chain peaks of high-altitude (**A**) and low-altitude *E. argus* (**B**) were eluted from the C4 RP-HPLC column, respectively. Five globin chain peaks were eluted for both high-altitude (**C**) and low-altitude *E. multiocellata* (**D**). The corresponding molecular weight and the inferred globin based on the molecular weight are shown beside the peaks.

**Figure 4 animals-14-01440-f004:**
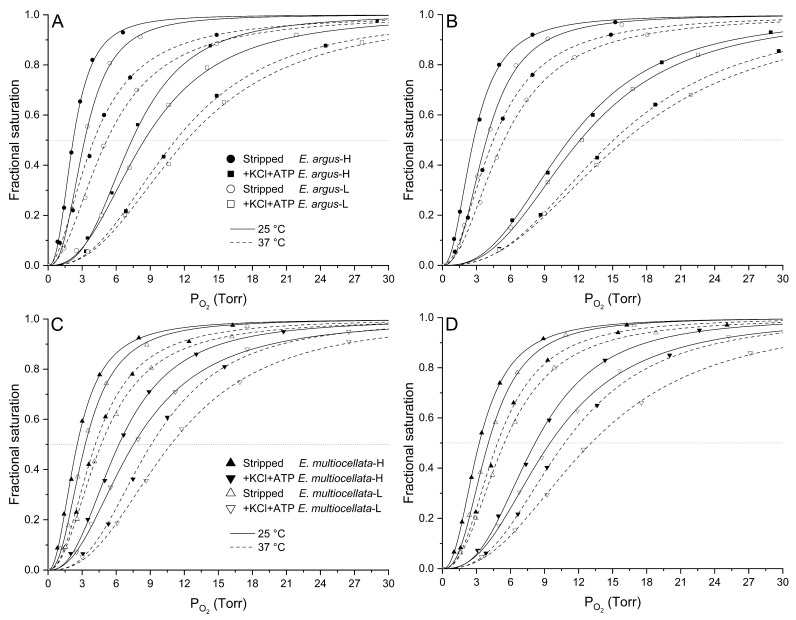
O_2_ equilibrium curves of high- and low-altitude *E. argus* Hbs ((**A**): pH7.8, (**B**): pH7.4) and high- and low-altitude *E. multiocellata* Hbs ((**C**): pH7.8, (**D**): pH7.4) at 25 °C (continued lines) and 37 °C (dotted lines) in the absence (stripped) and presence of Cl^−^ (added as 0.1 mol/L of KCl) and ATP (100-fold molar excess over tetrameric Hbs). In *E. argus*, the stripped status and presence of Cl^−^ and ATP were denoted by circles and squares, respectively. In *E. multiocellata*, the stripped status and presence of Cl^−^ and ATP were indicated by triangles and upside-down triangles. The high- and low-altitude populations are represented by solid and open symbols, respectively. *n* = 6 lizards for each population, and 6 technical repeats were performed for each experimental condition.

**Figure 5 animals-14-01440-f005:**
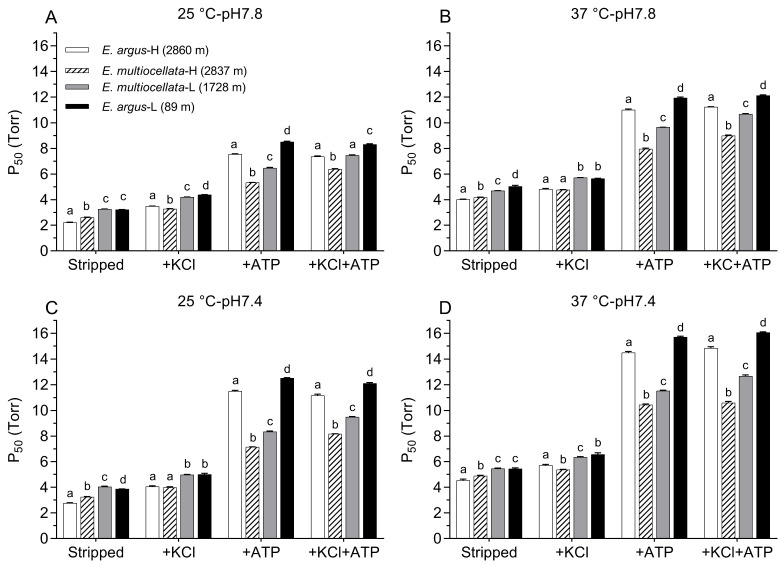
P_50_ values (mean ± SE) of Hbs from high- and low-altitude populations of *E. argus* and *E. multiocellata* measured at 25 °C pH7.8 (**A**), 37 °C pH7.8 (**B**), 25 °C pH7.4 (**C**), and 37 °C pH7.4 (**D**) in the absence (stripped) and presence of Cl^−^ (added as 0.1 mol l^−^ of KCl) and/or ATP (100-fold molar excess over tetrameric Hbs). Different letters above the P_50_ values of the four populations indicate significant differences between each pair of populations under the same allosteric condition according to one-way ANOVA using the LSD method of post hoc multiple comparisons (*p* < 0.05). *n* = 6 lizards for each population, and 6 technical repeats were performed for each experimental condition.

**Figure 6 animals-14-01440-f006:**
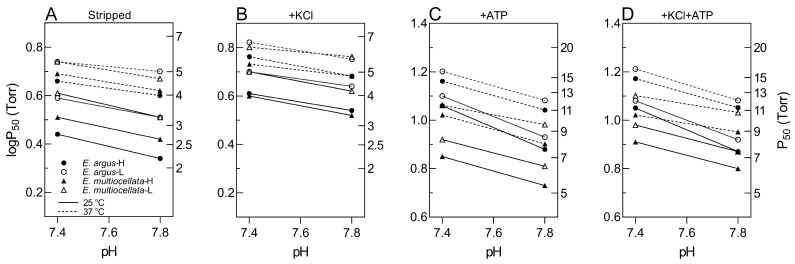
Bohr plots (logP_50_ vs. pH) of high- and low-altitude *E. argus* and *E. multiocellata* in the absence ((**A**): stripped) and presence of KCl (**B**), ATP (**C**), and +KCl+ATP (**D**) at 25 °C (continuous lines) and 37 °C (dotted lines). *E. argus* and *E. multiocellata* were denoted by circles and triangles, respectively. The high- and low-altitude populations are represented by solid and open symbols, respectively. The slope of these linear plots is equal to the corresponding Bohr factor (Φ, Table 2), which was used to indicate the magnitude of the Bohr effect. *n* = 6 lizards for each population and 6 technical repeats were performed for each experimental condition.

**Figure 7 animals-14-01440-f007:**
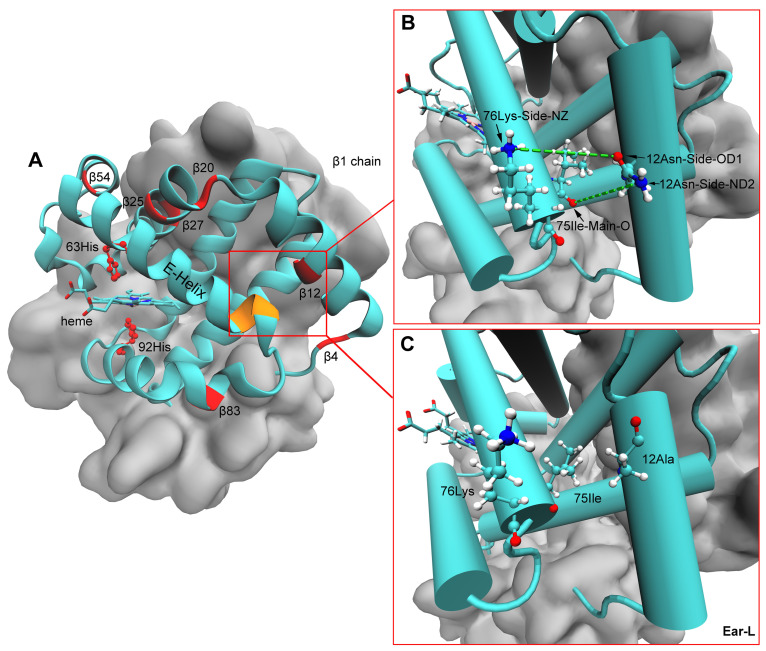
Spatial locations of heme, distal 63 His, and proximal 92 His on the β subunit of Hb models (**A**). The distal 63 His and proximal 92 His (represented by ball-and-stick) and the seven different amino acid residues in the β2 globin of the low-altitude *E. argus* compared with those of the other three *Eremias* lizards were marked in red, and the 75Ile and 76Lys were marked in yellow in (**A**). Hydrogen bonds (indicated by the green dotted line) formed between 12β2 and 75Ile and/or 76Lys (represented by ball-and-stick) on the adjacent E-helix in all Hb models composed of β2 globin with all α globins in high-altitude *E. argus* and high- and low-altitude *E. multiocellata* Hbs (**B**). However, these two hydrogen bonds were lost in the Hbs of low-altitude *E. argus* (Ear-L) due to the replacement of Asn with Ala at β12 (**C**). The β1 chain is represented by cyan, and the β2 and two α chains are represented by silver in the QuickSurf drawing method of VMD.

**Table 1 animals-14-01440-t001:** Anionic cofactor sensitivities of high- and low-altitude *E. argus* and *E. multiocellata* Hbs under various experimental conditions.

Cofactor Sensitivities	*E. argus*-H				*E. argus*-L				*E. multiocellata*-H				*E. multiocellata*-L			
Temperature	25 °C		37 °C		25 °C		37 °C		25 °C		37 °C		25 °C		37 °C	
pH	7.80	7.40	7.80	7.40	7.80	7.40	7.80	7.40	7.80	7.40	7.80	7.40	7.80	7.40	7.80	7.40
∆logP_50_																
KCl-stripped	0.20	0.17	0.08	0.10	0.13	0.11	0.05	0.08	0.10	0.09	0.06	0.04	0.11	0.09	0.09	0.07
ATP-stripped	0.53	0.62	0.44	0.50	0.42	0.51	0.37	0.46	0.31	0.34	0.28	0.33	0.30	0.31	0.31	0.32
(KCl + ATP)-stripped	0.52	0.60	0.45	0.51	0.41	0.49	0.38	0.47	0.39	0.40	0.33	0.34	0.36	0.37	0.36	0.37

**Table 2 animals-14-01440-t002:** Bohr factors for high- and low-altitude *E. argus* and *E. multiocellata* Hbs at 25 °C and 37 °C in the absence (stripped) and presence of Cl^−^ (0.1 mol/L of KCl) and/or ATP (100-fold molar excess over tetrameric Hbs).

	*E. argus*-H		*E. argus*-L		*E. multiocellata*-H		*E. multiocellata*-L	
Temperature	25 °C	37 °C	25 °C	37 °C	25 °C	37 °C	25 °C	37 °C
Bohr factor (Φ)								
Stripped	−0.25	−0.13	−0.20	−0.08	−0.23	−0.17	−0.23	−0.16
+KCl	−0.17	−0.18	−0.15	−0.17	−0.21	−0.13	−0.18	−0.12
+ATP	−0.46	−0.30	−0.42	−0.30	−0.31	−0.29	−0.28	−0.19
+KCl+ATP	−0.45	−0.30	−0.41	−0.30	−0.27	−0.18	−0.26	−0.19

**Table 3 animals-14-01440-t003:** Temperature effects (reflected by the overall change in enthalpy for oxygenation ΔH) on the O_2_ affinities of high- and low-altitude *E. argus* and *E. multiocellata* Hbs at pH7.8 and pH7.4 in the absence (stripped) and presence of Cl^−^ (added as 0.1 mol/L of KCl) and/or ATP (100-fold molar excess over tetrameric Hbs).

	*E. argus*-H		*E. argus*-L		*E. multiocellata*-H		*E. multiocellata*-L	
pH	7.8	7.4	7.8	7.4	7.8	7.4	7.8	7.4
ΔH (kJ/mol O_2_)								
Stripped	−25.74	−19.02	−16.06	−9.21	−17.27	−13.63	−10.64	−6.53
+KCl	−8.22	−9.18	−3.58	−4.79	−11.47	−6.67	−6.97	−2.96
+ATP	−11.64	−2.14	−9.13	−1.96	−12.97	−11.65	−13.02	−8.11
+KCl+ATP	−14.52	−5.66	−11.70	−5.57	−9.41	−4.12	−10.33	−5.97

## Data Availability

The original contributions presented in the study are included in the article. Further inquiries can be directed to the corresponding authors.

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
