# Peer review of "Convergent High O2 Affinity but Distinct ATP-Mediated Allosteric Regulation of Hemoglobins in Oviparous and Viviparous Eremias Lizards from the Qinghai-Tibet Plateau"

_animals, 2024, doi:10.3390/ani14101440_

Round 1
Reviewer 1 Report
Comments and Suggestions for Authors
This is a very interesting and meaningful study on the mechanism by which reptilian hemoglobin adapts to high altitude, which involved comparing two species of lizards from the Qinghai-Tibet Plateau with similar lizards from lower altitudes through genome analysis. The comparison of reptiles with mammals and birds in the discussion section is also interesting. The experimental methods used in this study are well thought out and the text is well written. I would like you to consider the following minor points.
l The Simple summary, unlike the Abstract, should be written in such a way that it is easier to understand for non-specialists to read. Instructions for authors say this should be written for a lay audience, i.e., no technical terms without explanations.
l If records of the highest elevations of high-altitude mammals were presented in the Introduction, it would be easier to compare them with the highest recorded altitude (5,300 m) inhabited by reptiles, and to understand that reptiles also have a highly adaptive capacity.
l According to the statistical analysis section (lines 268 and 269), the data are presented as mean ± SE. However, the data in Tables 1, 2, and 3, and Figure 4 appear to be presented as means only.
l Figures S1 (Lines 131, 517) and S2 (Ls 299, 437); Tables S1 (L 148), S2 (L 448), S3 (L 305), and S4 (Ls 334, 336): Figure and table numbers indicating supplemental data are used as well as the usual figure and table numbers in the text. Including only a few supplementary files is unproblematic, but if there are many such files, they should be compressed.
l The unit of "mol/L" instead of "moll-1" would be better in this journal. The use of "-1" in other units is similar.
Reviewer 2 Report
Comments and Suggestions for Authors
The manuscript “Convergent high O2 affinity but distinct ATP-mediated allo-2 steric regulation of hemoglobins in oviparous and viviparous Eremias lizards from the Qinghai-Tibet Plateau” detected the oxygenation properties and underlying mechanisms of Hbs in two Eremias lizards from the QTP compared with their low-altitude counterparts. This study found lizards from high-altitude could evolve Hbs with convergent high O2 affinity but distinct allosteric regulation, providing important information for understanding the adaptation evolution of high-altitude reptiles. The article is well written and the results are interpreted appropriately. However, some additional information is required to clarify the manuscript, and some grammar issues need to be improved.
Line 86“lack of physiological thermoregulation” as many reptiles can maintain their body temperatures in a narrow range through behavioral thermoregulation.
Line 92 “lack of physiological thermoregulatory capacity”
Line 219-222 Why do you choose 25°C as one experimental temperature?
Line 267-268 One-way ANOVA may be inappropriate as “species” may be an important factor affecting P50 values.
Line 332-343 No statistical values of the comparison of P50 among populations (F values) were provided.
Line 470 “performance ” may be “preference”.
Comments on the Quality of English Language
Some grammar issues need to be improved.
Line 65 add “and” between “mammals” and “birds”
Line 163-169 The tense is inconsistent.
